# Experimental and molecular predictions of the adjuvanticity of snail mucin on hepatitis B vaccine in albino mice

Parker Elijah Joshua[1], Cynthia Ogochukwu Nwauzor[1], Damian Chukwu Odimegwu[2‡], Uzochukwu Gospel Ukachukwu[1‡], Rita Onyekachukwu Asomadu[1‡], Timothy Prince Chidike Ezeorba[1]*

1 Faculty of Biological Sciences, Department of Biochemistry, University of Nigeria, Nsukka, Nigeria,
2 Department of Pharmaceutical Microbiology & Biotechnology, University of Nigeria, Nsukka, Nigeria

☉ These authors contributed equally to this work.
‡ These authors also contributed equally to this work
* timothy.ezeorba@unn.edu.ng

**Data Availability Statement:** All relevant data are within the paper and its Supporting Information files.

## Abstract

Although aluminum-containing adjuvants are widely used in human vaccination due to their excellent safety profile, they exhibit low effectiveness with many recombinant antigens. This study investigated the adjuvanticity of snail mucin with recombinant Hepatitis B Vaccine (rHBsAg). Twenty-five (25) female mice distributed unbiasedly into 5 groups were used in the study and were administered different rHBsAg/Mucin formulation at 7 days intervals. Blood samples were collected a day following the administration for analysis. The results of liver function and body weight analysis were indications that snail mucin had no adverse effect on the mice. The treatment group (administer mucin and rHBsAg) showed significantly (P<0.05) higher mean titres of anti-HBsAg antibodies when compared with the negative controls and the positive control administered with two doses of rHBsAg after the boost doses (day 28). Furthermore, a comparable immune response to positive control administered with three doses rHBaAG was recorded. *In silico* prediction, studies of the protein-protein interaction of a homology modelled snail mucus protein and HBsAg gave an indication of enhanced HBV antigen-antibody interaction. Therefore, this study has shown that snail mucin possesses some adjuvant properties and enhances immune response towards rHBsAg vaccine. However, there is a need for further molecular dynamics studies to understand its mechanism of action.

## Introduction

Hepatitis B Virus (HBV) is a causative agent of hepatitis B infection, Approximately, a million people die annually from HBV-related chronic liver diseases, such as liver failure, liver cirrhosis and hepatocellular carcinoma (HCC) [1]. The disease is majorly transmitted across human population through unprotected sex, birth transmission, transfusion of contaminated blood, and the use of object that are contaminated [2].

**Funding:** The author(s) received no specific funding for this work.

**Competing interests:** The authors have declared that no competing interests exist

Vaccination is the most effective measure to decrease the worldwide HBV prevalence and its complications [3]. Generally, vaccination is aimed to induce protective immunity against a unique epitope of an antigen, and in some vaccines, this can be enhanced by addition of adjuvants. Adjuvants are substances added to vaccines to improve the immune response of pure antigens, which may not stimulate adequate immune response on their own [4].

For several decades, aluminium containing adjuvants (alums) have been very effective in human vaccination and were generally the only approved adjuvants by the United States Food and Drug Administration (FDA) due to its' excellent track record of safety, low cost and its application with variety of antigens. However, in the modern era of recombinant proteins and small peptides vaccination, alums are implicated with a number of limitations such as local reactions, zero effectiveness to some recombinant antigens, and poor augmentation ability to some cell-mediated immune responses, such as cytotoxic T-cell responses [5]. Hence, there are several ongoing studies to investigate other effective adjuvants such as biodegradable polymeric particle technology.

The discovery of a safe and effective HBV vaccine derived from yeast-derived recombinant hepatitis B surface antigen (rHBsAg) promises a global reduction in HBV incidence. However, to generate the effective immune responses, individuals need about three or more doses of the vaccine after a couple of months interval [6]. Consequently, maintaining a consistent re-immunization rate for multiple-administration program is quite difficult, especially in developing countries [7]. Hence, this study is birthed from the need for the development of more effective adjuvant/vaccine delivery systems against HBV, possibly requiring only a single round of immunization to yield long-lasting immune responses.

Many recent studies on adjuvant development for hepatitis B vaccines are now focusing on the use of biodegradable polymeric particles (BPP) for adjuvants and have reported promising results [1]. Snail mucin being a natural mucoadhesive polymer can be classified as a BPP owing to its biocompatibility, non-antigenic/non-toxic nature as well as biodegradability in the living system [8]. Many recent studies have discovered snail mucin to possess wound healing and age renewing properties [9]. Other studies further reported that snail mucin possesses diverse biophysical and pharmaceutical applications and are effective for mucoadhesive drug delivery agent [10]. In this study, the suitability of snail mucin polymer as an adjuvant for recombinant HBsAg vaccine was investigated in albino mice.

## Materials and methods

### Animals

The Faculty of Biological Sciences, University of Nigeria, Board on Ethical Clearance for Animal Research granted approval to perform the research on Albino mice. Female albino mice aged 6 to 8 weeks were used for the study. The animals were obtained from the Animal house of the Faculty of Veterinary Medicine, University of Nigeria and were used for the experimental study. They were acclimatized for 10 days under standard environmental conditions with a 12-hour light/dark cycle, fed with standard pellets (Pfizer Livestock feeds Plc, Enugu, Nigeria) and tap water was given ad libitum. Precaution were taken to ensure that the blood collection procedure did not result to the loss of the animal sight, thereby affecting their feeding and other physiological activities.

### Reagents

All chemicals and reagents used for the study were of analytical grade and included 2 M $H_2SO_4$ (JHD), 96% Acetone (JHD), Alanine aminotransferase (ALT) Kit (DiaLab, Austria), Aspartate aminotransferase (AST) Kit (DiaLab, Austria), Carbonate-Bicarbonate Buffer (pH

9.5), Citrate Phosphate Buffer (pH 5.0), DMSO (JHD, China), methanol, Fat-free Milk powder (Dano-Slim, Nigeria), Hydrogen Peroxide (JHD, China), Phosphate Buffered Saline (pH 7.4), Leishman stain, Tetramethylbenzidine (TMB) Tablets–(SIGMA, USA), Tween 20 (Germany).

## Vaccine and secondary antibodies

Recombinant hepatitis B surface antigen vaccine, Euvax-B (LG Chemical, South Korea) was used as the vaccine candidate for immunization. Goat anti-mouse horseradish peroxidase-conjugated IgG, IgG1 and IgG2a immunoglobulins (Southern Biotechnology, USA) were used as secondary antibodies. Both vaccine and secondary antibodies were stored at 4˚C until needed.

## Snails

Adult giant African land snails (*Archachatina marginata*) were procured from the Orie-Orba market, Enugu, Nigeria. A total number of 130 snails were used and were identified in the Department of Zoology and Environmental Biology, University of Nigeria.

## Extraction of snail mucin

The study adopted the method published by Kalu et al. [11] to extract the mucin from the giant African land snail.

## Preparation of snail mucin dispersion (adjuvant)

Snail mucin adjuvant was prepared according to the method of Adikwu and Nnamani [12]. A quantity (0.01g) of powdered snail mucin extract was weighed using an electronic balance and dispersed in 1 ml of normal saline. The dispersion was stirred for 15 minutes and allowed to hydrate completely for 24 hours before use. Thus, the concentration of the snail mucin was 1% w/v.

## Experimental design

A total of 25 animals were randomly divided into five (5) groups of five (5) animals each as follows:

**Group 1 (3DVac):** First positive control (rHBsAg vaccine alone: 3 doses)

**Group 2 (2DVac):** Second positive control (rHBsAg vaccine alone: 2 doses)

**Group 3 (2DVacSM):** First treatment group (rHBsAg vaccine + Snail mucin: 2 doses)

**Group 4 (2DSM):** Second treatment group (Snail mucin alone: 2 doses).

**Group 5 (3NS):** Negative control (Normal saline: 3 doses)

## Immunization protocol

Euvax-B (LG-Chemical, South Korea) was employed as a vaccine candidate. The mice were injected intramuscularly in a prime-boosting fashion with either 100μL (2μg) of rHBsAg vaccine alone (3 and 2 doses respectively), 100 μL of rHBsAg vaccine + 50 μL of snail mucin, 50 μL of snail mucin alone, or 100 μL of normal saline. The animals were vaccinated on days 1 and 15 (for 2 doses) and on days 1, 15 and 22 (for 3 doses) by a well-trained researcher in animal handling. Extensive efforts were deployed to minimize animal suffering during does administration and animal handling.

An aliquot of 0.5 mL of blood samples were collected from all experimental mice on days 0, 14, 21 and 28 before immunizations through the retro-orbital plexus and centrifuged at 5000 rpm for 10 minutes to obtain clear sera which were frozen until needed for immunochemical analyses.

## Determination of body weights and liver weight

The mice were weighed daily throughout the study duration (from day 0 to day 28) using an electronic weighing scale. On day 28, the mice were euthanized by the method of cervical dislocation and their livers were harvested and weighed using an electronic weighing scale. The protocol of euthanasia by cervical dislocation was performed quickly by trained researcher, to ensure painless animal sacrifice. No anaesthesia or analgesia administer to the animal, to prevent interference with other histopathology analysis.

## Differential white blood cell count

The mice differential white blood cell count was determined on days 0, 14, 21 and 28 using the method described by Carr and Rodak [13].

## Liver function test of mice immunized with snail mucin-adjuvanted rHBsAg vaccine

Serum AST and ALT concentrations were determined using DiaLab diagnostic kits (Wiener Neudorf, Austria). Determination of AST and ALT activities were based on the method described by Thomas [14], Moss & Henderson [15]. An abnormal increase in the amount of AST and ALT in the blood above the normal threshold is an indication of liver disease or damaged hepatic cells. Principally, serum Aspartate Aminotransferase (AST) and Serum Alanine Aminotransferase (ALT) activity were determined quantitatively by its enzymatic effect on the oxidation of NADH to $NAD^+$.

## Measurement of antibody titres of IgG, IgG1 and IgG2a against rHBsAg using indirect Enzyme-Linked Immunosorbent Assay (iELISA)

Analysis of specific total antibodies (IgG) were performed using an optimized indirect ELISA method described by Nejati *et al*. [16] on a 96-well ELISA plates (BRANDplates®, Germany) coated with 100 μL of rHBsAg. Similarly, the specific IgG1 and IgG2a subclasses were performed using goat anti- mouse IgG1 and IgG2a secondary antibodies (Southern biotechnology, USA).

## Statistical analysis

The results were expressed as mean ± standard deviation (S.D). Data were analyzed using SPSS using one-way and two-way ANOVA and subjected to Duncan tests. Differences between the mean of the treated groups and control would be considered significant at $P < 0.05$.

## Homology modelling of mucus protein and model validation

Structure prediction through homology modelling of the already deposited amino acid sequence of snail mucus protein on the NCBI protein database (Accession: QEG59314.1) was adopted to investigate the nature of the protein-protein interaction between the vaccine and the antigen [17,18]. The crystal structure of Mucus Proteins in garden snails was not available in the Protein Data Bank (PDB - www.rcsb.org), therefore, the 340 amino acids sequence of

the mucus protein was subjected to 3D structure prediction using homology modelling strategy. A PDB blast analysis was carried out for the selection of model templates for the 340 amino acid sequence of snail mucus (Accession—QEG59313.1). The protein showing the highest sequence identity and query coverage was selected from the PDB-blast analysis and was used as a template for the model building of 3D structure of snail mucus protein on Swiss Model (https://swissmodel.expasy.org/interactive/) [19].

The predicted 3-D models were further refined with the GMQE (Global Model Quality Estimation) score assessment method in open sources Swiss Model package. The quality of the modelled structure with respect to stereo-chemical geometry and energy were analysed by SAVES v6.0 (https://saves.mbi.ucla.edu/) protein quality check package. The PDB structures of the snail mucus were also used for further protein interaction studies and in silico docking and dynamic studies.

## Homology modelling of Hepatitis B surface antigen (HBsAg) and model validation

The complete crystal 3D structure of the envelope protein of Hepatitis B has not been deposited in the PDB (PDB - www.rcsb.org) although Li et al. [20] crystalized a partial sequence of the envelope protein known as Pres1. This study adopted homology modelling to obtain a model structure for the 385 amino acids sequence of the complete envelope protein of HBV, downloaded from the National Centre for Biotechnology Information (NCBI) protein database (www.ncbi.nlm.nih.gov/), having an accession number of AAK94656.1. The obtained model structure of HBsAg was also subjected to quality check on the SAVES v6.0 (https://saves.mbi.ucla.edu/) protein quality check package. The model structure was hence used for molecular docking studies and other dynamics *in silico* experiments.

## Protein-protein alignment of snail mucus protein and HBsAg and generating structural morphs with PyMOL 2.4 version 2.4.1

Computational protein morphing produces an intermediate conformation resulting from the transitioning of protein conformations during protein-protein interaction. In this study, we used Morphing plugin in PyMOL 2.4 version 2.4.1 to generate a tentative morph structure, when a simulated interaction between the modelled HBsAg structure interacts with the modelled snail mucus protein. The generated morph was used for the docking experiment.

## Studies of antigen (HBsAg)–antibody interaction using molecular docking

The modelled HBsAg 3D structure was docked to both the immunoglobulin structure against the antigen (PDB_ID– 5YAX) and the human immunoglobulin against HBV (PDB_ID– 4QGT). Similarly, the PDB file of the generated morph when snail mucus and antigen interact, was also docked with HBV antibody (PDB_ID– 5YAX) and human immunoglobulin (PDB_ID– 4QGT). The docking result was compared for both the modelled surface antigen (HBsAg) and the Morph structure.

## Results

### Snail mucin-adjuvanted rHBsAg vaccine had no adverse effects on experimental mice

To investigate if Snail Mucin as an adjuvant could pose some adverse effects on the animals, the body weight of the mice was measured throughout the study (day 0 to day 28). All five groups of mice had comparable growth patterns, inferring that there was no side effect caused

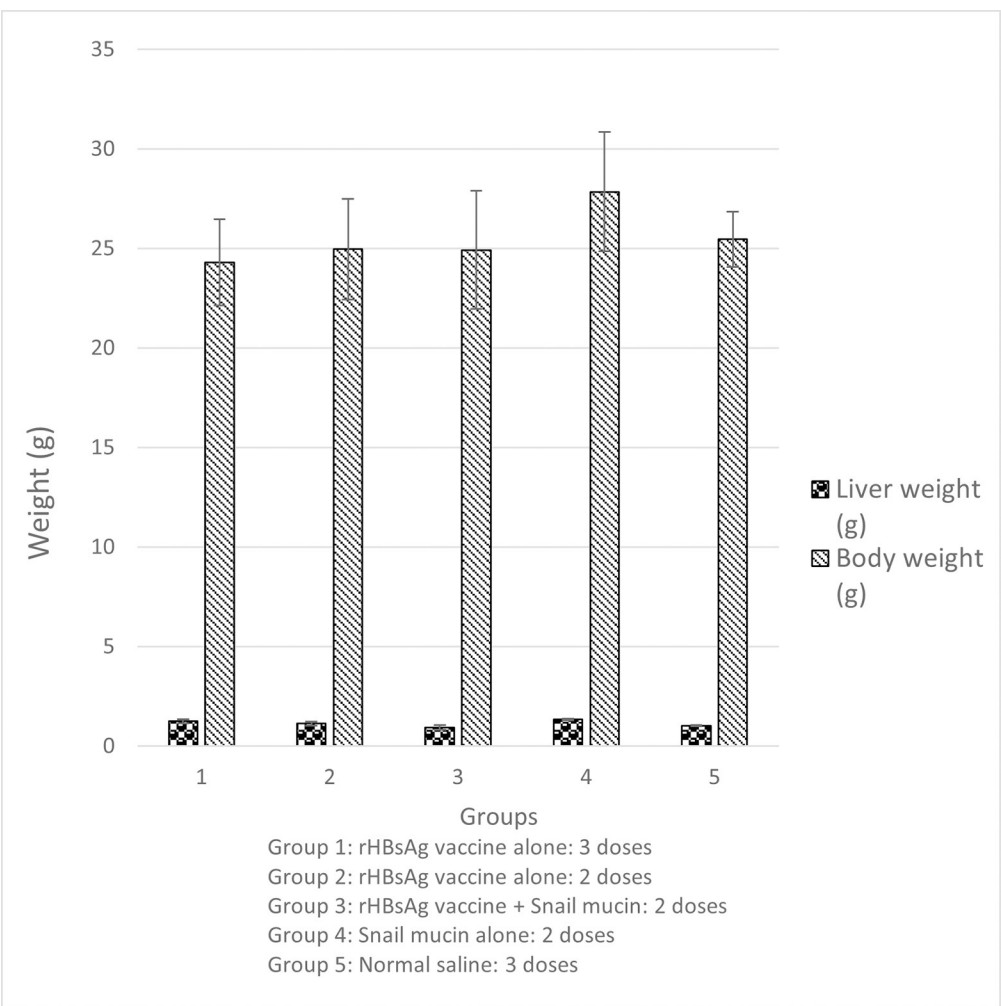

**Fig 1. Effect of snail mucin-adjuvanted rHBsAg vaccine on the body and liver weight of experimental mice.**
Results are expressed as mean ± SD (n = 5) with a confidence interval of 0.05.

by the administration of snail mucin or co-administration of snail mucin and vaccine on the animals (Fig 1).

Moreover, from the physical examination of livers, in all groups, the external surface of the liver was smooth, shiny, and reddish brown in colour suggesting healthy organs and slight variation in the liver weights was recorded among the groups (Fig 1). In summary, there was no mortality recorded in any experimental group throughout the experiment.

## Snail mucin-adjuvanted rHBsAg vaccine had no significant toxicity in the liver

To evaluate if any form of liver toxicity did result from administering the Snail Mucin-Adjuvanted rHBsAg Vaccine. The activities of the liver-marker enzymes were measured. In the result shown in Table 1, the AST activity of the 2DVacSM experimental group was non-significantly ($p > 0.05$) higher when compared to other experimental groups and the controls. Moreover, the value was within the normal range of AST for mice (54–298 U/L).

Table 1. Effect of snail mucin-adjuvanted rHBsAg vaccine on liver marker enzyme activities of experimental mice.

| Group | AST (U/L) | ALT (U/L) |
|---|---|---|
| Group 1 | 147.99 ± 16.09[a] | 47.62 ± 5.18[a] |
| Group 2 | 157.01 ± 13.49[a] | 41.71 ± 3.54[a] |
| Group 3 | 160.92 ± 18.80[a] | 42.20 ± 4.92[a] |
| Group 4 | 135.01 ± 12.05[a] | 45.58 ± 2.94[a] |
| Group 5 | 140.51 ± 15.21[a] | 40.35 ± 2.27[a] |

Results are expressed as mean ± SD (n = 3). Mean values with different letters as superscripts along the columns were considered significant at $p < 0.05$.

Group 1: rHBsAg vaccine alone: 3 doses.

Group 2: rHBsAg vaccine alone: 2 doses.

Group 3: rHBsAg vaccine + Snail mucin: 2 doses.

Group 4: Snail mucin alone: 2 doses.

Group 5: Normal saline: 3 doses.

Similarly, the ALT activity of the 2DVacSM and 2DSM experimental group fell with the normal range of ALT enzyme activities for mice (17–77 U/L) and showed non-significantly ($p > 0.05$) difference when compared to other groups and the control (Table 1).

## Snail mucin-adjuvanted rHBsAg vaccine had no significant effect on the granulocyte count of experimental mice

Granulocytes consist of the Neutrophils, Eosinophils and Basophils. To observe the effect of Snail Mucin adjuvants on granulocytes, a differential white blood cell count was performed on day 0, 14, 21 and 28. The results from Fig 2 showed a steady significant ($p < 0.05$) increase in the neutrophil count from day 0 to day 28 which was comparable among all experimental groups.

Furthermore, there were no significant ($p > 0.05$) differences in the monocytes, eosinophils and basophils count among all experimental groups throughout the experiments (result not shown). Hence, there were no special effects caused by snail mucin on the neutrophils, monocytes, and eosinophils.

## Effect of snail mucin-adjuvanted rHBsAg vaccine on cellular immune response (lymphocyte count) of experimental mice

To observe the effect of Snail Mucin-Adjuvanted rHBsAg Vaccine on lymphocytes, a differential white blood cell count was performed. There was a significant ($p < 0.05$) decrease in Lymphocyte count for all experiment groups from day 0 to day 28 (Fig 3).

## Effect of snail mucin-adjuvanted rHBsAg vaccine on the titre value of immunoglobulin G (IgG), immunoglobulin G subclass 1 (IgG1) and immunoglobulin G subclass 2a (IgG2a) of experimental mice

Indirect ELISA techniques were adopted to investigate the adjuvant effects of snail mucin on IgG, IgG1 and IgG2a titre. From the result as shown in Fig 4A, there was a significant ($p < 0.05$) increase in the mean IgG titre of Group 3, co-administered with snail mucin and vaccine on day 28 (0.51±0.07) compared to before immunization (day 0–0.19±0.08), after prime immunization (day 14–0.19±0.02) and after boost immunization (day 21–0.27±0.03).

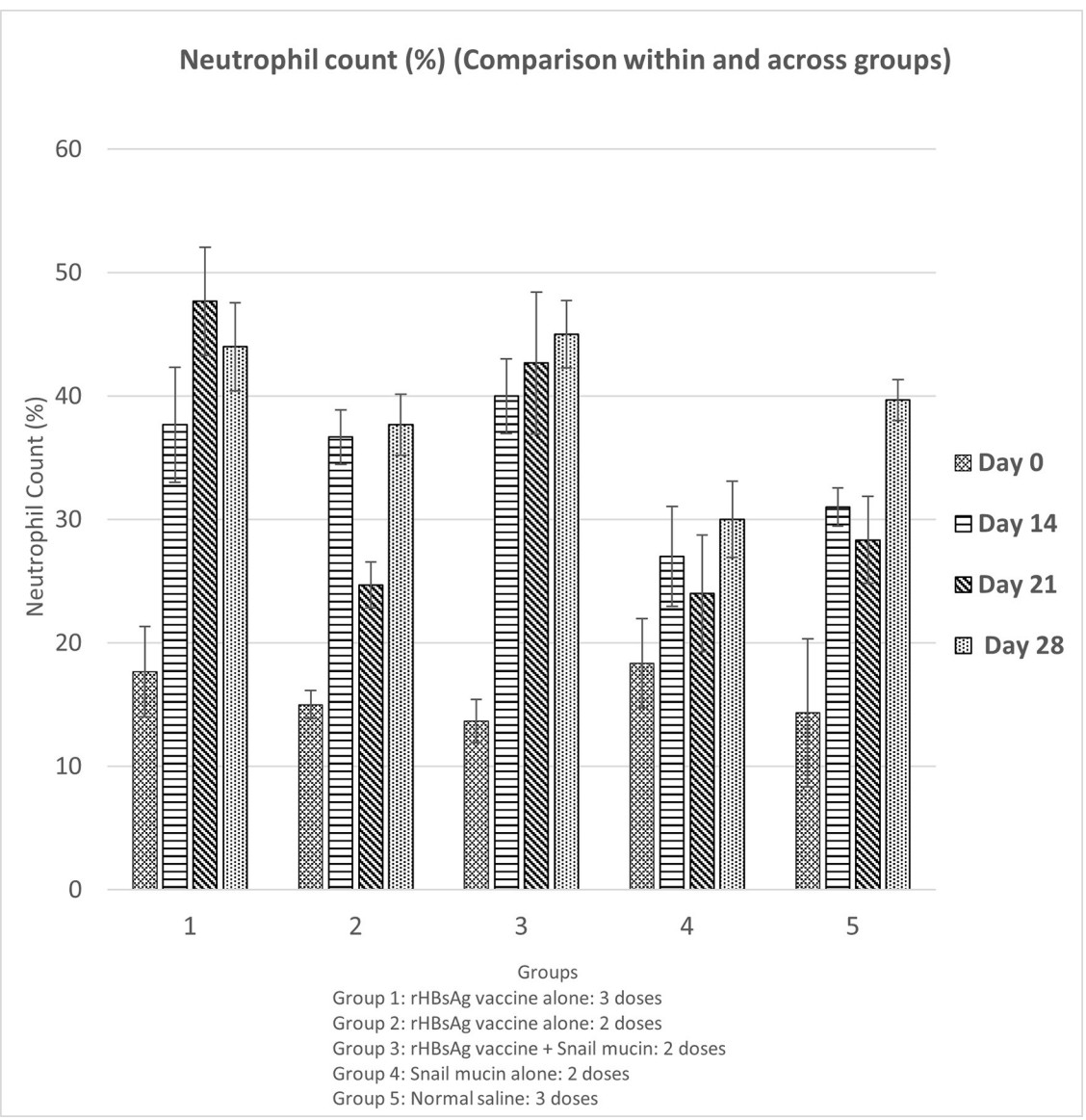

**Fig 2. Effect of snail mucin-adjuvanted rHBsAg vaccine on neutrophil count of experimental mice.** Results are expressed as mean ± SD (n = 5) with a confidence interval of 0.05.

Moreover, on day 28, the IgG titre of the group co-administered snail mucin and vaccine (Group 3) was significantly ($p<0.05$) higher than the negative controls (group 4 and 5 –administered with only mucin and normal saline) and the positive control (group 2 –administered with 2 doses of snail mucin).

Furthermore, there was a significant ($p<0.05$) increase in the mean IgG1 titre of the group co-administered snail mucin and vaccine on day 21 (0.46± 0.02) and day 28 (0.73±0.03) after boost immunization compared to day 0 (0.32±0.10) and day 14 (0.20±0.01), before and after prime immunization. Moreover, on day 28, the mean IgG1 titre of mice co-administered snail mucin and vaccine was found to be comparable to the groups administered 3 and 2 doses of vaccine alone and significantly ($p<0.05$) higher than the negative control (Fig 4B).

On the other hand, the titre results of the IgG2a from Fig 4C showed a contrasting trend when compared with IgG and subclass IgG1. There was a progressive and significant ($p<0.05$)

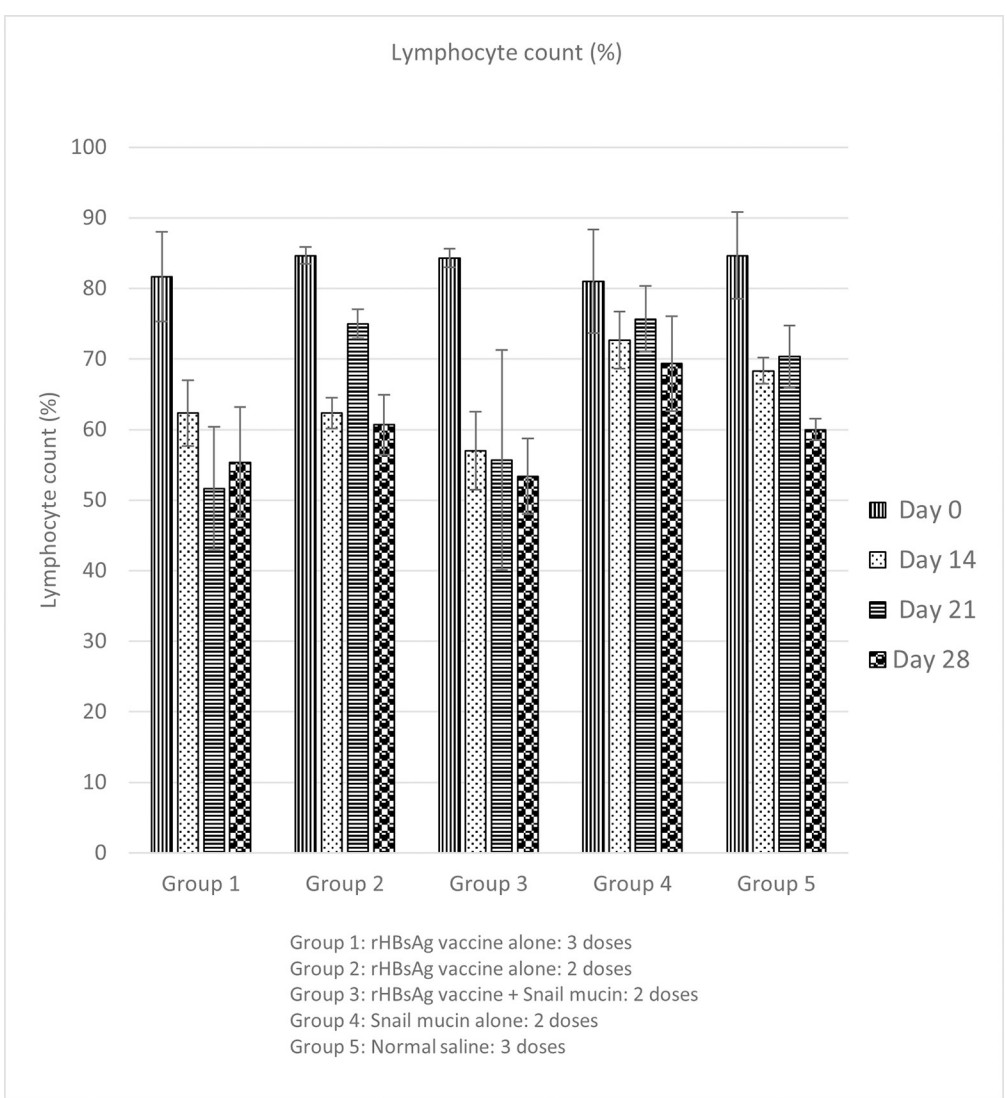

**Fig 3. Effect of snail mucin-adjuvanted rHBsAg vaccine on lymphocyte count of experimental mice.** Results are expressed as mean ± SD (n = 5) with a confidence interval of 0.05.

decrease in mean IgG2a titre of the group co-administered snail mucin and vaccine after prime and boost immunizations [day 14 (0.13±0.03), day 21 (0.11±0.02) and day 28 (0.11 ±0.02)] compared to before immunization [day 0 (0.26±0.11)]. More so, the IgG2a titre of group 3 was comparable to the negative controls (group 4 and 5) and significantly lower (p<0.05) that the positive control.

## Homology modelling of snail mucus protein and HBV envelope protein

Performing a sequence blast of snail mucus protein against the protein data bank, the best structural homolog being the structure of proximal thread matrix protein 1 (PTMP1) from the mussel byssus, was selected to generate the homologous model of the snail mucus protein.

The homology modelled snail mucus structure (Fig 5A) was interesting as it showed 8 Alpha helixes and 8 beta sheets. The local quality score for most of the residue was on average above 0.7 and the about 92.58% of the residues were favoured by the Ramachandran

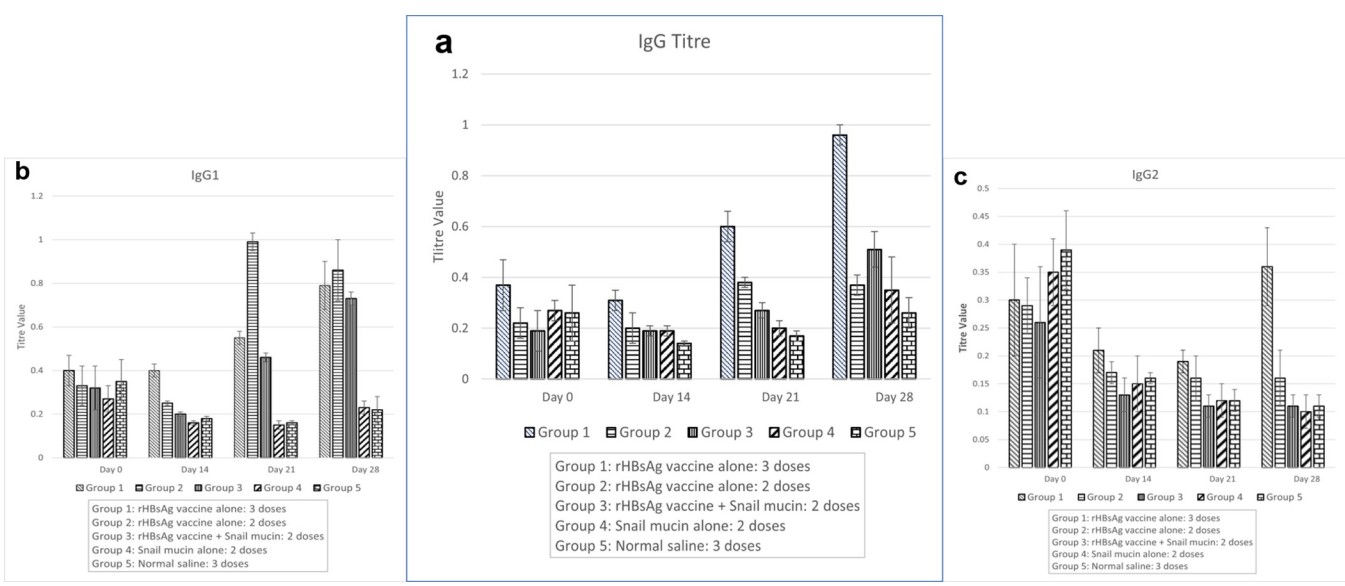

**Fig 4. Effect of snail mucin-adjuvanted rHBsAg vaccine on IgG, IgG1 and IgG2a titre of experimental mice.** The figure shows the IgG titres (a), IgG1 titre (b) and IgG2a titre (c) across the experimental groups obtain from an iELISA experiment. Results are expressed as mean ± SD (n = 5) with a confidence interval of 0.05.

simulation. Finally, the modelled 3D structure of the protein gave a QMEAN score of above 0.5 with a standard deviation of the mean |Z-score| between 1 and 2.

Similarly, the structure of HBsAg (Fig 6A) was obtained through Homology Modelling of the amino acid sequence downloaded from the NCBI website (accession number—AAK94656.1) also on Swiss-Model (https://swissmodel.expasy.org/). A prior blast of the amino acid sequence against the Protein Data bank (PDB) gave a low query cover of 15%, but however, a 100% identity to the crystal structure (PDB ID - 5YAX_C) of a human neutralizing antibody against Hepatitis B virus, bound to a HBV preS1 peptide. On further analysis of the 5YAX crystal structure, the 100% identity was a result of the bounded Pres1 peptide.

### Rheomorphic structural state from interaction of snail mucus protein and modelled HBsAg did not give a better docking score than the standalone HBsAg

A structural morph with about 30 rheomorphic states was generated from the alignment of modelled HBsAg and snail mucus protein using PyMOL v1.4.1. The generated morphs were docked to the immunoglobulin protein (PDB_ID– 5YAX) and the score was compared to the score from docking the standalone HBsAg to the same antibody (PDB_ID– 5YAX). The best predictions from the docking experiment, the morph, gave a lower negative score of -2646.06. Meanwhile, the best prediction from docking the standalone HBsAg gave a higher negative score of -4296.27. Although the score from the Morph docking was not better than the standalone antigen, it was discovered that both the Morph and the crystal Pres1 antigen interacted with the similar variable region for the Fab Fragment of the antibody (Fig 7), while the best prediction of the full HBsAg did not bind to the either variable region of the Fab fragments.

### Discussion

The toxicity profile and high cost of currently available vaccine adjuvants have necessitated the search for safer and more efficacious adjuvants for vaccines. Although the level of

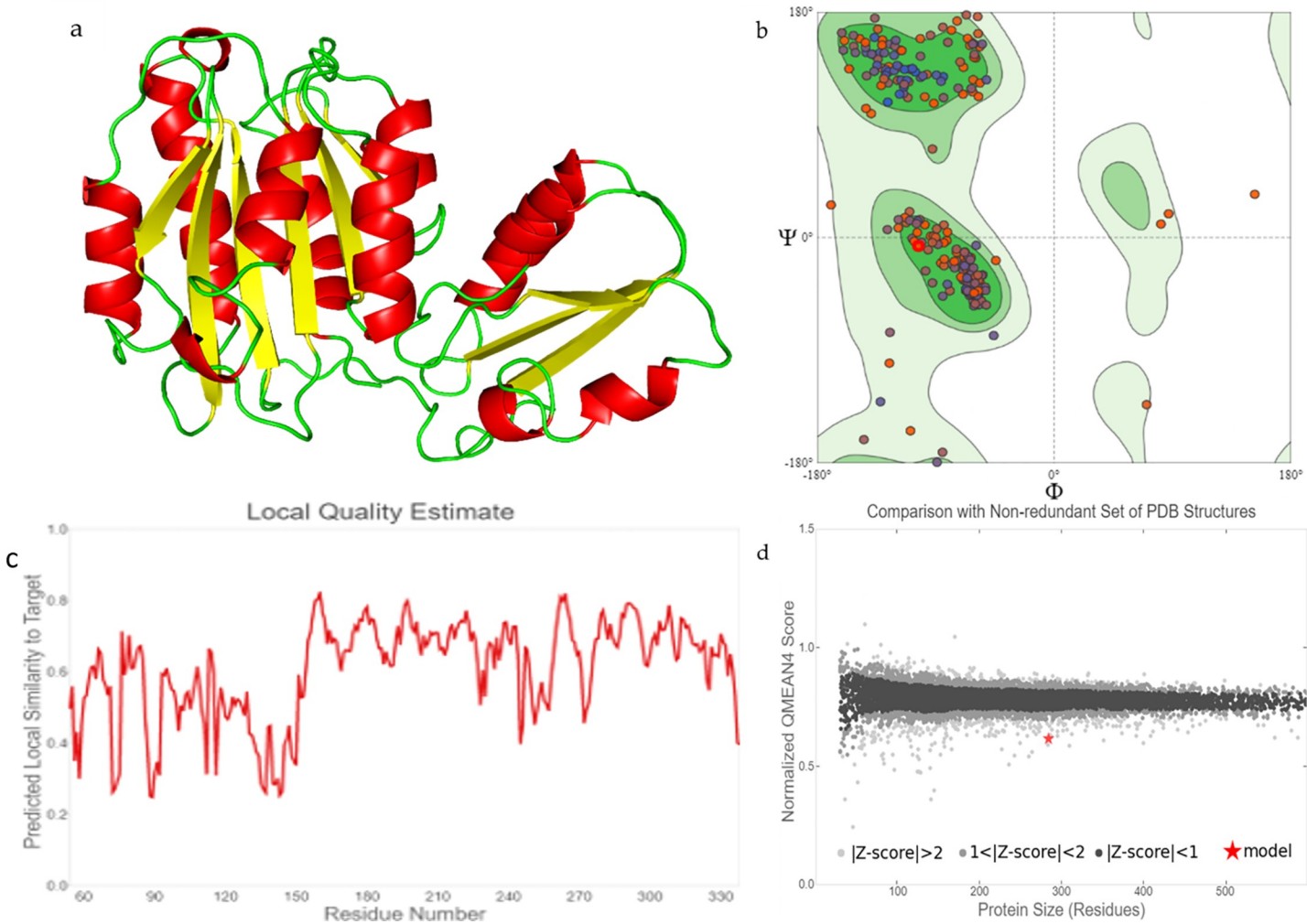

**Fig 5. Homology modelled structure of snail mucus protein [Cornu aspersum] with accession number QEG59313.1.** The Ramachandran modelled favoured about 92.58% residue and with a few outliers. The local quality score was on average above 0.7 for most of the residue. Finally, when in comparison with other non-redundant PDB structure, the model structure gave a ZMEAN of above 0.5 and a deviation of between 1 and 2.

biotechnological advancement for generating vaccines have grew greatly with numerous recorded successes, there are no measurable development in the science of adjuvant, that may greatly enhance antibody responses to several bacterial and viral infection. The currently widely used hepatitis B vaccine antigen is a highly purified and derived from yeast-base recombinant technology. According to Schijns *et al.* [21], recombinant or purified vaccine antigens have poor immunogenicity due to removal of pathogen associated molecular patterns (PAMPs) which serve as intrinsic adjuvants; hence, they need other factors and components to boost their protective capacity against infections. These additional components are called adjuvants (from a Latin word "adjuvare" which means to help) are needed to enhance the immunogenicity of vaccine antigens improving the cumulative titre of antibody and effector T cells [21].

In the present study, vaccination of mice with recombinant hepatitis B surface antigen vaccine combined with snail mucin as well as with snail mucin alone did not result in growth retardation of the experimental animals. Experimental animals showed no difference in weight gain patterns compared to the controls. Although there were observed declines in body weights of all the animals after immunization and blood collection protocols because of

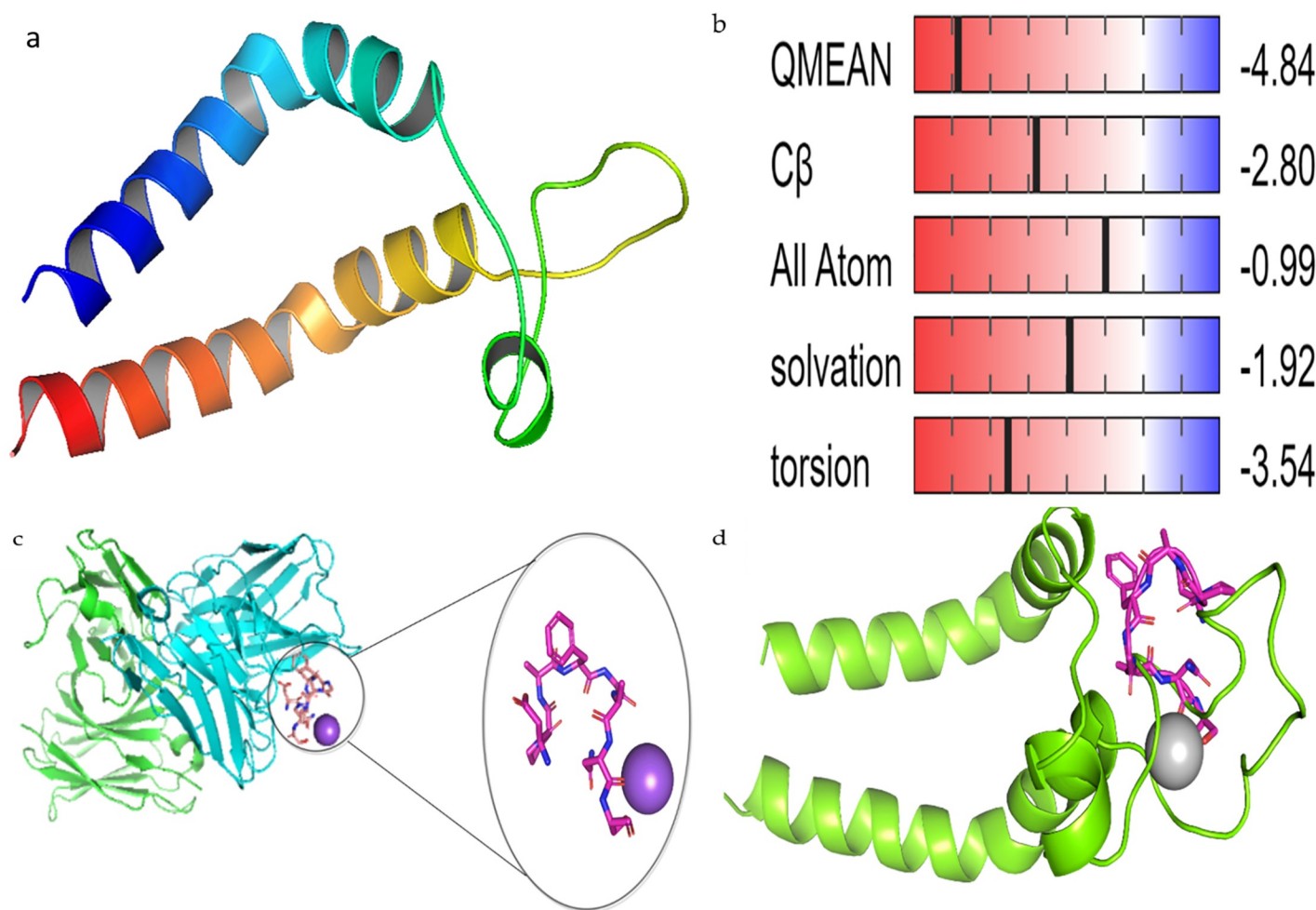

**Fig 6.** a. Homology Modelled Structure of complete envelope protein of HBV showing a form of Helix-Turn-Helix conformation. b) Quality check checked of the modeled structure with regards to other non-redundant structures deposited in the protein data bank. c) X-ray crystal structure of a human neutralizing antibody bound to an HBV preS1 peptide, which was zoomed out to show the structure of the peptide. d) Aligned structure of the modelled HBV envelope and crystal structure of Pres1 peptide.

handling stress, the animals recovered within a day or two. The observed normal growth pattern suggests that the treatments were safe.

Organ weight data are very useful for evaluation of toxicological change as a positive increase in liver weight can be accurately correlated with hepatocellular hypertrophy. To ensure proper experimental comparison and inference, Lazic, *et al*. [22] recommends that organ weight data are expressed as an organ-to-body weight ratio across the experimental groups. The mean liver weight of the group co-administered snail mucin and vaccine was comparable to the groups administered 3 and 2 doses of vaccine alone and the group administered normal saline. However, the liver tissues from the group administered snail mucin alone showed a significant increase in the mean weight as compared to the group administered normal saline. This increase in weight can be explained based on organ-to-body weight ratio. A higher body weight corresponds to a higher organ weight and vice versa. The mean body weight of the group administered snail mucin alone was higher than any of the other groups and so it is expected that the group should have the highest mean liver weight. Hence, there were no observed toxicological changes (hepatic hypertrophy) induced in the animals by

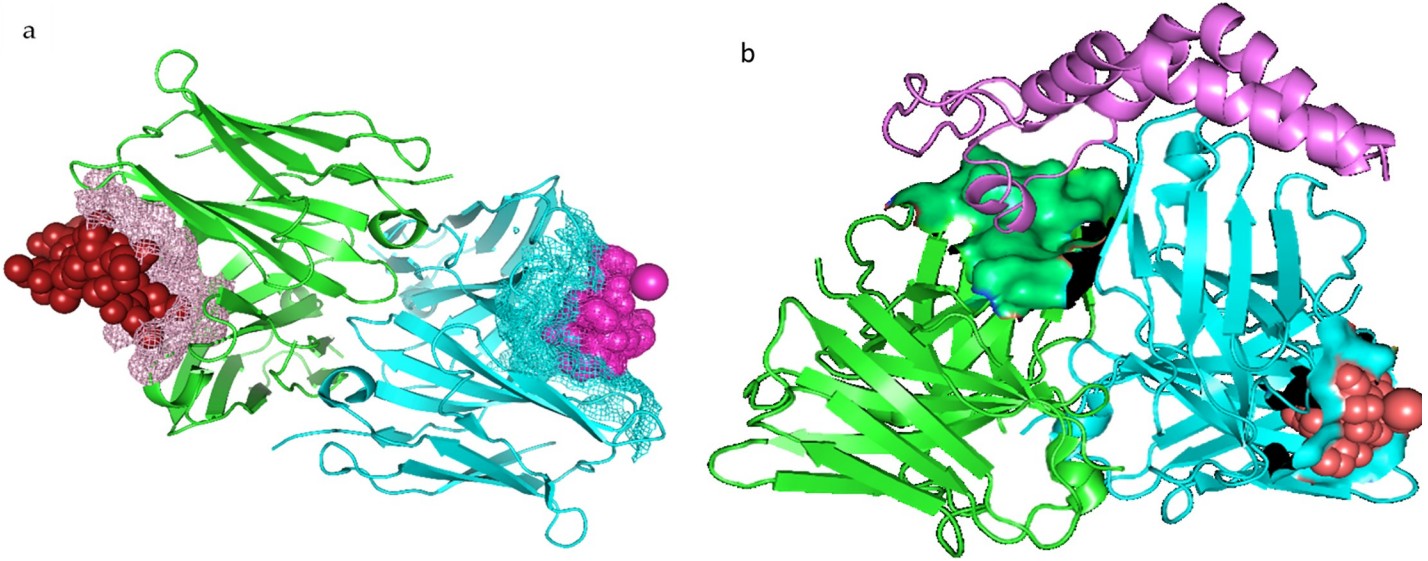

**Fig 7. Docking morph originating from the interaction of the HBsAg and snail mucus protein with the human immunoglobulin against HBV (PDB_ID– 5YAX) and comparison with the result when standalone HBsAg is docked to 5YAX.** a). The best prediction (-2646.06) for docking Morph to 5YAX. The brick red sphere represents the morph while the purple mesh are the surrounding interacting atoms on the variable region on the Fab segment of the heavy chain. The purple sphere represents the Pres1 peptide interacting with similar variable region of the light chain (blue mesh) of the Fab fragment. b). The best prediction when the standalone HBsAg envelope protein (the purple Helix-Turn-Helix Structure) is docked 5YAX. The green surface are residues of 5YAX interacting with the HBsAg which are not exactly the variable region of the heavy chain where antigen can bind. The red sphere is the Pres1 peptide interacting with residues on the light chain of 5YAX (blue surface).

immunization with snail mucin, rHBsAg vaccine or snail mucin-rHBsAg vaccine. This is further supported by the external morphological features of the organs which suggested that the organs were healthy. In all the groups, the external surface of the liver was smooth, shiny, and reddish brown in colour. These are the characteristics of normal, healthy livers. Similarly, Yuan *et al.* [23] adopted the use of the body weight of chickens to evaluate if any toxicity arose from the use of sunflower seed oil and ginseng stem-leaf saponins when used as an adjuvant for Newcastle disease vaccine.

To further ascertain that there was no hepatotoxicity caused by the vaccine formulation, serum transaminase levels were investigated. Following onset of liver enlargement, the increase in hepatic-derived enzymes (transaminases, alkaline phosphatase, and γ-glutamyltransferase) are frequently observed [24]. Alanine Aminotransferase (ALT) and Aspartate Aminotransferase (AST) activities are vital pathology parameters helpful in the investigation of adverse effects of drugs/chemicals on the liver. Increases in serum ALT activity are thought to be due to damage to, and leakage from hepatocyte cell membranes, resulting in a release of enzymes from the cytosol into the blood [25]. Increases in ALT activity of two to three folds are indicative of hepatocellular damage. The AST and ALT levels in the group co-administered snail mucin and vaccine were found to be comparable to the negative control group. They were within the normal acceptable range for mice (17–77 U/L for ALT and 54–298 U/L for AST). Hence, there was no indication of hepatotoxicity caused by the vaccine formulation.

Recent empirical data suggest that many adjuvants enhance T and B cell responses by interacting with several components of the innate immune system, rather than by direct effects on the lymphocytes themselves [26]. This was the rationale behind the differential white blood cell count performed in the present study. Four of the five white blood cell types identified during the differential WBC count (neutrophils, monocytes, eosinophils, and basophils) are components of the innate immune system.

The increase in the monocyte count of the group co-administered snail mucin and vaccine compared to the groups administered 3 and 2 doses of vaccine alone on day 14 is evidence of immune stimulatory capacity of the formulation. Monocytes, which differentiate into macrophages and dendritic cells upon maturity, are highly important in activating adaptive immune responses. They are highly important in phagocytosis, antigen presentation, cytokine production, and antimicrobial and cytotoxic activities [27]. Thus, stimulation of monocytes is an indication that snail-mucin adjuvanted rHBsAg vaccine is more immunogenic than the vaccine alone.

No significant basophil release was detected in any of the groups. This is in line with O'Connell et al. [28], who reported that basophils are rarely present in murine peripheral blood. The absence of basophils in the group co-administered snail mucin and vaccine proves that the vaccine formulation did not produce any allergic reaction.

There was an upregulation of eosinophils in the group co-administered snail mucin and vaccine on day 21 compared to the groups administered 3 and 2 doses of vaccine alone. Although eosinophils are more important in resistance to parasites, they are phagocytic cells, and their upregulation suggests innate immune system activation. According to Iwasaki and Medzhitov [27], eosinophils are important to the innate response and help to link the innate immune response to the adaptive immune response.

Neutrophils are largest in number of all other leukocytes and functions as the critical effector cells of the innate immune system [29]. Neutrophils expresses a broad pattern of recognition receptors such as toll-like receptors, Fc receptors and complement components. Majorly, neutrophil annihilate microorganisms through phagocytosis, release of cytotoxic granules and use of neutrophil extracellular traps [30] and hence, they are important innate response that coordinate to improve adaptive immune responses [31]. Several studies have proven the ability of mouse neutrophils to serve as antigen presenting cells (APCs) or influence the capacity of professional APCs to present antigens [32]. Vono et al. [32] showed that neutrophils can present cognate antigens to antigen-specific memory CD4+ T cells. And hence, they are invaluable in the immunity against disease.

Co-administration of snail mucin with vaccine caused significant increases in neutrophil counts compared to administration of vaccine alone (Fig 2). There was a corresponding decrease in lymphocyte count. This is in consonance with Provencher et al. [33] who stated that in mice, lymphocyte counts decrease as neutrophil counts increase. The observed increase in neutrophil count indicates that neutrophils probably functioned as APCs, leading to increased activation of antigen-specific B lymphocytes as observed from the data on humoral immune responses.

The mean IgG titre of the group co-administered snail mucin and vaccine showed a non-significant ($p > 0.05$) increase on day 14 after prime immunization. This could be explained by the fact that adaptive immune responses are activated after boost immunization. After the boost immunization, there was also a non-significant ($p > 0.05$) increase in the mean IgG titre observed on day 21. This non-significant increase could be attributed to the possible entrapment of the vaccine antigen by snail mucin microparticles. Polymeric biodegradable microparticles used as vaccine adjuvants have been shown to entrap antigen and release it gradually over a period. The advantage of this mechanism is that it mimics the prime and boosts doses of vaccination, and therefore elicits robust immune responses in a single administration [34].

The delivery of antigens loaded on polymeric particle systems has several advantages: it prolongs antigen presence, enhances DC-mediated antigen uptake, directs stimulation of DCs, and promotes longer and better immune responses [35]. The proposed antigen entrapment mechanism was further supported by a marked significant ($p > 0.05$) increase in the mean IgG titre of the group co-administered snail mucin and vaccine compared to the group

administered 2 doses of the vaccine alone on day 28 (Fig 4A). However, there was a decrease (non-significant) in the mean IgG titre of the group administered 2 doses of the vaccine alone (group 2) on day 28, suggesting antigen clearance and termination of immune stimulation. Snail mucin serves as a biodegradable polymer which generally have great application in medicines. They are used to deliver drug and vaccine into the body are they are enzymatically hydrolysed in the body releasing their content [36].

The significant increases in the mean IgG1 titres of the group co-administered snail mucin and vaccine on days 21 and 28 compared to day 14 (Fig 4) can be explained by entrapment and sustained release of antigen by adjuvant (snail mucin), resulting in continuous and persistent stimulation of the immune system over a period of time such that by day 28, the mean IgG1 titre of the group co-administered snail mucin and vaccine (2 doses) was comparable to the group administered 3 doses of vaccine alone. The increased IgG1 titre indicates a strong Th2 type (humoral) response.

Cytotoxic T-lymphocytes (CTLs) stimulated by Th1 response is not essential for protective immunity against HBV [37]. IgG2a stimulates Th1 responses while IgG1 stimulates Th2 responses. This is a possible explanation for the trend of results obtained for mean IgG2a titre (Fig 4). The group co-administered snail mucin and vaccine had comparable IgG2a titres with groups administered 2 doses of vaccine alone, snail mucin alone and normal saline. Another possible explanation for this observation is that the strain of mice used for the study may not be expressing IgG2a. Shinde et al. [38] reported that in some strains of mice there is no expression of IgG2a; instead these mice express the novel IgG2c. Thus, it can be inferred that IgG2a is not a useful immunoglobulin subclass to monitor immune responses to recombinant hepatitis B surface antigen vaccine in non-IgG2a expressing mice.

The homology modelled snail mucus structure (Fig 5A) and the complete envelope protein of the HBV (HBsAg) was modelled on Swiss-Model program using the downloaded amino acid sequence from the NCBI protein database (Accession number—AAK94656.1). Although, the complete envelope structure has not been deposited in the protein data bank, one study conducted by Li et al. [20] successfully elucidated the X-ray crystal structure of a peptide which is part of the envelope protein of the HBV and responsible for virulence (Pres1 peptide). The structure of the Pres1 peptide was deposited on the PDB database when bonded to a human neutralizing antibody against HBV infection [20]. Pres 1 peptide is part of the envelope protein that is responsible for the virulence of HBV. Pres1 binds to the Sodium Taurocholate Cotransporting Polypeptide (NTCP) of the hepatocytes, hence resulting in Hepatitis [22].

Results from the docking experiment showed the standalone protein had a better docking score than morph when the best prediction out of 100 was compared. However, when the structure of the best prediction was observed closely with PyMOL v2.4.1 and VMD (Fig 7A and 7B), the morph interacted with mainly with the variable region of the Fab fragment of 5YAX while the standalone protein interacted with other residues other than the variable region. The antibody (immunoglobulin) has a consistent structure constituting two the heavy chain and two light chains, which are further grouped into the two Fab (fragment of antigen binding) and the one Fc (Fragment crystallizable) [39]. The specificity of antibody to a single antigen is brought about by the variable segment of the Fab region. So, if the interaction between the standalone antigen and the antibody occurs around residues away from the variable Fab segment, it implies that there might be an interaction which is not a favourable antigen-antibody complex. Interaction of the standalone HBsAg with snail mucus generated several rheomorphic structures that interacted with the variable region of the Fab fragment, although with a lower docking score.

## Conclusion

Based on this study, snail mucin could serve as an effective and safe adjuvant for recombinant hepatitis B surface antigen vaccine. It is well-tolerated and elicits appropriate immune responses. It probably exerts its effects by sustained and gradual release of vaccine antigen thereby producing continuous and persistent stimulation of immune responses. The proposed mechanism mimics the prime and booster doses of recall immunizations and directly translates into reduction of the number of doses/injections needed to attain protective antibody levels. One promising benefit of such is single-administration immunization programs and cost effectiveness.

## Supporting information

**S1 Table. Effect of hepatitis-B vaccine modulated with Egg Shell membrane on eosinophil count.**
(PDF)

**S2 Table. Effect of snail mucin-adjuvanted rHBsAg vaccine on monocyte count of experimental mice.**
(PDF)

**S3 Table. Effect of snail mucin-adjuvanted rHBsAg vaccine on basophil count of experimental mice.**
(PDF)

## Author Contributions

**Conceptualization:** Parker Elijah Joshua, Damian Chukwu Odimegwu.

**Data curation:** Timothy Prince Chidike Ezeorba.

**Formal analysis:** Uzochukwu Gospel Ukachukwu, Rita Onyekachukwu Asomadu, Timothy Prince Chidike Ezeorba.

**Investigation:** Cynthia Ogochukwu Nwauzor.

**Methodology:** Cynthia Ogochukwu Nwauzor, Uzochukwu Gospel Ukachukwu, Timothy Prince Chidike Ezeorba.

**Project administration:** Parker Elijah Joshua, Uzochukwu Gospel Ukachukwu.

**Resources:** Parker Elijah Joshua.

**Software:** Rita Onyekachukwu Asomadu, Timothy Prince Chidike Ezeorba.

**Supervision:** Parker Elijah Joshua, Damian Chukwu Odimegwu.

**Validation:** Parker Elijah Joshua, Damian Chukwu Odimegwu.

**Visualization:** Timothy Prince Chidike Ezeorba.

**Writing – original draft:** Cynthia Ogochukwu Nwauzor.

**Writing – review & editing:** Rita Onyekachukwu Asomadu, Timothy Prince Chidike Ezeorba.

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
