## [Decision Letter · Decision Letter 0]

1 Apr 2021

PONE-D-21-02861

Experimental and Molecular Predictions of the Adjuvanticity of Snail Mucin on Hepatitis B Vaccine in Albino Mice

PLOS ONE

Dear Dr. Ezeorba,

Thank you for submitting your manuscript to PLOS ONE. After careful consideration, we feel that it has merit but does not fully meet PLOS ONE’s publication criteria as it currently stands. Therefore, we invite you to submit a revised version of the manuscript that addresses the points raised during the review process.

In particular the Figure 4 results that is pivotal in the paper's conclusions and statements. Methodological approaches are not sound and well described.

We look forward to receiving your revised manuscript.

Kind regards,

Isabelle Chemin, PhD

Academic Editor

PLOS ONE

Journal Requirements:

In your Methods section, please state the volume of the blood samples collected for use in your study.

To comply with PLOS ONE submissions requirements, in your Methods section, please provide additional information on the animal research and ensure you have included details on (1) methods of sacrifice, (2) methods of anesthesia and/or analgesia, and (3) efforts to alleviate suffering.

Reviewers' comments:

Reviewer's Responses to Questions

**Comments to the Author**

1. Is the manuscript technically sound, and do the data support the conclusions?

Reviewer #1: No

2. Has the statistical analysis been performed appropriately and rigorously? 

Reviewer #1: No

3. Have the authors made all data underlying the findings in their manuscript fully available?

Reviewer #1: Yes

4. Is the manuscript presented in an intelligible fashion and written in standard English?

Reviewer #1: Yes

5. Review Comments to the Author

Reviewer #1: The current manuscript authored by Joshua et al., focuses the study of adjuvanticity of snail mucin with recombinant Hepatitis B Vaccine 28 (rHBsAg). The authors state that this study has shown that snail mucin possesses some adjuvant properties and enhances immune response towards rHBsAg vaccine. This is an interesting report. However, the data is not convincing, especially the data in Figure 4. The conclusions drawn are not supported by the experimental data. This manuscript in its present form does not meet the Plos One publication standard. I would ask the authors to address several key issues for further consideration.

Major concerns:

1. In figure 4, which is the most important figure in this manuscript, the authors tried to make a statement that treatment group (administer mucin and rHBsAg) showed significantly (P<0.05) higher mean titres of anti-HBsAg antibodies when compared with the negative controls and the positive control administered with two doses of rHBsAg. This statement is not true and is not based on the experiment result. Group 3 could be significantly higher than Group 5, the negative controls. However, the most important comparison is between Group 2 ( rHBsAg vaccine alone: 2 doses) and Group 3 ( rHBsAg vaccine + Snail mucin: 2 doses). There was no time point that showed that Group 3 is significantly better than Group 2, in terms of IgG, IgG1 or IgG2 titres.

2. The authors should consider demonstrating IgG, IgG1 or IgG2 of anti-HBsAg antibodies in Figure 4 in separate panels. For example, one panel shows IgG titer only in all time point. It is crucial to add statistics results (error bars) to each of the figures.

3. Snail mucin toxicity needs to be determined by a dosage test. After the animals are treated with increasing amount of Snail mucin, the body weight of the mice will be measured daily to determine the minimum amount of Snail mucin that causes adverse effects on the animals.

4. A survival test is strongly recommended to support the conclusion. Lethal dose (LD50 is recommended) of Hepatitis B will be given to all groups of the mice after they received their treatment on Day 28. If the mucin is helpful as adjuvants, we will find more mice in Group 3 survive the lethal virus challenge than the mice in Group 2.

6. PLOS authors have the option to publish the peer review history of their article (what does this mean?). If published, this will include your full peer review and any attached files.

Reviewer #1: No

---

## [Author Response · Author response to Decision Letter 0]

27 May 2021

Dr. Isabelle Chemin, 

Academic Editor,

PLOS ONE Journal. 

Dear Dr. Isabelle, 

RESPONSE TO ACADEMIC EDITOR AND REVIEWER

I sincerely appreciate the privilege of reviewing our research article entitled “Experimental and Molecular Predictions of the Adjuvanticity of Snail Mucin on Hepatitis B Vaccine in Albino Mice”. Thank you very much for your comments and feedback, and also that of the reviewer. I have thoroughly effected all possible corrections and made relevant additions based on your feedback. I sincerely hope you find them meritorious for your reputable Journal (PLOS ONE).

Based on the Academic Editor comments which focused on the journal requirements, here are my responses and corrections that have been effected on the manuscripts 

1. I ensured that the manuscript now meets the PLOS ONE’s style requirements, especially with regards to proper figure and table labelling, using the right font size for level 1 and 2 headings and using upper cases only for the first letter of all headings.

2. In my first submission, I erroneously omitted the volume of blood samples used for the study in the method section. However, in this revised manuscript, the volume of blood collected, which was 0.5 mL, has been included in the method section.

3. Finally, the methods on handling the animals have been explicitly included in the revised manuscript. The method of cervical dislocation was adopted for animal euthanasia by a well-trained researcher, who ensured that the suffering of the animals were reduced to the barest minimum. We did not administer anaesthesia or analgesia to the animals despite their pain-relieving activities, to avoid interference with other intended histopathology analysis. More so, snail mucin has been reported to possess some anaegestic properties, hence possibly reducing the pain in the animals (Adikwu and Nnamani, 2005). 

Based on the peer-reviewer’s major concerns, here are my responses and corrections that have been effected on the manuscripts. 

Major concerns 1 and 2: 

I sincerely appreciate all the comments of the reviewer.

Fig 4 from my initial submission was clumsy. It was a composite graph showing the IgG, IgG1 and IgG2 titre values together, for different time point. Hence, the figure did not explicitly despite some of the claims from my results.

To address this issue based on the reviewer suggestion, I have separated the composite graphs into three separate graphs. The new graphs (Fig 4a, 4b and 4c) now show the mean titre value for IgG, IgG1 and IgG2 with their various statistics (error bar) respectively. 

The reviewer pointed out that my claim of observing a significantly (P<0.05) higher mean titre of anti-HBsAg bodies in group 3 (administered with mucin and rHBsAg) when compared to negative control (group 5) was correct. However, my result didn’t support the assertion of a high antibody titre in group 3 when compared to the positive control (group 2-adminstered with rHBsAg only) at any time point. 

The new Fig 4a showed that on day 28, a significantly higher IgG titre value of the experimental group 3 (administered with 2 doses of the vaccine and mucin), when compared with both the positive control group 2 (administered only with 2 doses of rHBsAg vaccine only) and the negative control group 4 and 5 (administered with only mucin and normal saline respectively).

Although, the result on day 14 and 21 did not show significant (P>0.05) differences in the IgG titre across the treatment group 3 and control group 2 & 5, as compared to what was observed on day 28. This result patterned our thinking to the possible mechanisms of adjuvanticity of snail mucin on rHBsAg, as was explained in the discussion section.

We explained that the mucin, possibly acts as a Polymeric Biodegradable Microparticles, entrapping the antigen and releasing it gradually over a period of time. The proposed antigen entrapment mechanism was supported by a marked significant (p>0.05) increase in the mean IgG titre of group 3 (co-administered snail mucin and vaccine) compared to group 2 (administered 2 doses of the vaccine alone) on day 28 (Fig 4a).

Similarly, the new Fig 4b, showed a very interesting result. Although the IgG1 titre of the treatment group 3 was significantly lower than the positive control group 2 on day 21 and non-significantly lower on day 28, there was a relatively significant boost in the IgG1 titre for the treatment group from day 21 to day 28, while the positive control group 2 showed a decrease in IgG1 titre from day 21 to day 28. This further supports our assumptions of possible antigen entrapping mechanism of Snail Mucin. 

Lastly, the results of the new Fig 4c showed a contrasting trend for the mean titre value of IgG2. Studies have shown that Cytotoxic T-lymphocytes (CTLs) stimulated by Th1 response are not essential for protective immunity against HBV (Konda et al., 2013). It is known that IgG2a stimulates Th1 responses while IgG1 stimulates Th2 responses. Hence, the possible explanation for a low IgG2 titre across the group. 

Major concerns 3: 

The reviewer pointed out the need for toxicity studies which is very relevant for determining the effective dosage that can be accommodated in the animals.

In this present study, it is true we did not perform any toxicity studies on the mice, despite its significance. However, our choice for the dose formulation for snail mucin was not done blindly but was guided by a study conducted by Adikwu and Nnamani (2005). It was reported in that study that snail mucin was well tolerated, as the animals (mice 20-25 g) did not show any symptoms of toxicity and none of the animals died (Adikwu and Nnamani, 2005). Since, a similar experiment on the animals has been published in the past, we decided not to repeat the same experiments, with the aim of avoiding the unjustifiable use of animals for research - based on our institutional animal laws.

Moreover, our results from our study showed that snail mucin was non-toxic to the animals as, no animal died during the study, no significant changes in the weight of the animal and no obvious allergic reaction was observed at the point of administration. Finally, the liver weight and histology were similar for all the experimental groups. 

Major concerns 4: 

The reviewer recommended that a survival test study be conducted to fully support the conclusions that snail mucin is helpful as an adjuvant for rHBsAg vaccine.

It is true that we did not conduct a survival test, during the experimental phase of the studies. This study is relevant as it may give more confirmatory evidence to our assertion that snail mucin has some adjuvant properties in rHBsAg. However, our conclusion was based on general scientific knowledge about the correlation of elevated antibody titres with vaccine efficacy. Moreover, the survival test is imperative to determine whether these antibodies generated could really neutralize the virus.

The results we have presented have shown that snail mucin has some possible adjuvant properties, through the possible Polymeric Biodegradable entrapment mechanism – entrapping the antigen, releasing it in bits to sustained last longing immunity, and reducing the need for multiple vaccinations.

Despite the relevance of this suggested study (survival test), it may be very difficult to perform this experiment now as we have euthanized all the animals used for the studies. More so, culturing HBV for infecting the animal is very difficult due to restrictions within the institution when working with infectious viral particles. Regardless, this will be done in a future study, when we secure collaborations with a virology institute/laboratory.

Thank you very much for taking the time to go through my responses. Hope you find them worthy. 

Yours sincerely, 

Timothy P. C. Ezeorba 

Timothy.ezeorba@unn.edu.ng

---

## [Decision Letter · Decision Letter 1]

17 Jun 2021

PONE-D-21-02861R1

Experimental and Molecular Predictions of the Adjuvanticity of Snail Mucin on Hepatitis B Vaccine in Albino Mice

PLOS ONE

Dear Dr. Ezeorba

Thank you for submitting your manuscript to PLOS ONE. After careful consideration, we feel that it has merit but does not fully meet PLOS ONE’s publication criteria as it currently stands. Therefore, we invite you to submit a revised version of the manuscript that addresses the several minor points raised during the review process.

We look forward to receiving your revised manuscript.

Kind regards,

Isabelle Chemin, PhD

Academic Editor

PLOS ONE

Journal Requirements:

Reviewers' comments:

Reviewer's Responses to Questions

**Comments to the Author**

1. If the authors have adequately addressed your comments raised in a previous round of review and you feel that this manuscript is now acceptable for publication, you may indicate that here to bypass the “Comments to the Author” section, enter your conflict of interest statement in the “Confidential to Editor” section, and submit your "Accept" recommendation.

Reviewer #1: All comments have been addressed

2. Is the manuscript technically sound, and do the data support the conclusions?

Reviewer #1: Yes

3. Has the statistical analysis been performed appropriately and rigorously? 

Reviewer #1: Yes

4. Have the authors made all data underlying the findings in their manuscript fully available?

Reviewer #1: Yes

5. Is the manuscript presented in an intelligible fashion and written in standard English?

Reviewer #1: Yes

6. Review Comments to the Author

Reviewer #1: I want to thank the authors for re-submitting the article "Experimental and Molecular Predictions of the Adjuvanticity of Snail Mucin on Hepatitis B Vaccine in Albino Mice". The modifications made to this paper have greatly improved its quality.

I have 2 minor remarks for this manuscript:

- Can the authors explain why Group 2 is similar to Group 4 (Snail Mucin only, no vaccine) on Day 28 in Fig.4a?

-A figure legend usually has a title, and Materials and method (A description of the techniques used). P-values and the sample size, if applicable, should also be included. Please complete all figure legends in the manuscript.

7. PLOS authors have the option to publish the peer review history of their article (what does this mean?). If published, this will include your full peer review and any attached files.

Reviewer #1: No

---

## [Author Response · Author response to Decision Letter 1]

7 Jul 2021

University of Nigeria, Nsukka

Faculty of Biological Sciences

Department of Biochemistry

Timothy P. C. Ezeorba 24th May 2021

(Lecturer II) 

Dr. Isabelle Chemin, 

Academic Editor,

PLOS ONE Journal. 

Dear Dr. Isabelle, 

RESPONSE TO ACADEMIC EDITOR AND REVIEWER

Once again, I sincerely appreciate the privilege of reviewing our research article entitled “Experimental and Molecular Predictions of the Adjuvanticity of Snail Mucin on Hepatitis B Vaccine in Albino Mice”. Thank you very much for your comments and feedback, and also that of the reviewer. I sincerely hope you find this rebuttal meritorious for your reputable Journal (PLOS ONE).

Review based on the journal requirements

I have reviewed the reference list and ensured that they are now appropriate. I replaced Ref number [11], Adikwu and Ikejiuba, (2005) with a new reference (Kalu et al., 2019).

Reason - I discovered that the journal website where Adikwu and Ikejiuba, (2005) published have been pulled out and is no longer accessible. The new reference (Kalu et al., 2019) is more recent and addresses similar methods and ideas as the previous reference used.

To the best of my knowledge, other references are appropriate for publication.

Based on the peer-reviewer’s major concerns, here are my responses and corrections that have been effected on the manuscripts. 

I want to sincerely appreciate the reviewer for the detailed and thorough review of this work. I have ensured to respond appropriate to the reviewer’s comments to the best of my ability. Presented below are the reviewer’s comments and my responses.

- Can the authors explain why Group 2 is similar to Group 4 (Snail Mucin only, no vaccine) on Day 28 in Fig.4a?

Thank you very much for pointing that out from our result as depicted in Figure 4a. 

The results on Figure 4a show the IgG titre value measured using iELISA for all the experimental groups from day 0 to 28. The results on day 28 showed a similar IgG titre value for animals in group 2 (administered 2 doses of the vaccine alone) and group 4 (a negative control administered 2 doses of mucin alone), as pointed out by the reviewer. Intuitively, one may think that there should be a significant difference among the two groups, however, a non-significant (P>0.05) increase in group 2 as compared to group 4 was observed. The possible reasons for an increased IgG in the group 4 animals may be due to immunogenic responses in some of the animals within the group which may have been challenged with antigens within their environment (inhaled or ingested). This may be evident as pointed out by statistical analysis which showed a considerably higher error bar in group 4 on day 28, when compared to other experimental groups.

Furthermore, there are possibilities that snail mucin, since foreign to the animal, may have resulted in a level of antibody production, accounting for the IgG titre in group 4. Although there are very limited studies on the immunogenicity of snail mucins, few studies available highlighted contrasting opinions. A study conducted by Karne et al. (2017) showed mucin protein from Macrochlymus indica (a species of Terrestrial Land Snail) to possess some immunogenic properties after an IELISA experiment. While, another study conducted by Harti et al. (2018) pointed out that snail mucus from Achatina fulica on its own could not stimulate lymphocyte proliferation. Hence, there is need for future studies to investigate thoroughly the immunogenic potential of Snail mucin from different varieties of snails. 

- A figure legend usually has a title, and Materials and method (A description of the techniques used). P-values and the sample size, if applicable, should also be included. Please complete all figure legends in the manuscript.

Thank you very much for the comments. 

It is very true that a description for each of the figures are necessary. I have done that in my previous submission. The figure legends were place in the body of the article, at a position where the Figures should be placed. This was in line with the journal format for submission. 

However, I have improved the legends, by adding some of its vital components as you pointed out from your comment. Thank you. 

Thank you very much for taking the time to go through my responses. Hope you find them worthy. 

Your sincerely, 

Timothy, P. C. Ezeorba 

Timothy.ezeorba@unn.edu.ng

 

References 

Karne S. P., Gupta A., Sumesh S., kamble, S. and Bharat S. (2017). Exploration of mucin protein from Macrochlymus indica and determined its immunogenicity studies. Journal of Biomedical and Pharmaceutical Research, 6(1): 33-38. Retrieved from https://jbpr.in/index.php/jbpr/article/view/83

Harti, A. S., A. Murharyati, D. S. S, and M. Oktariani (2018). The effectiveness of snail mucus (Achatina fulica) and chitosan toward limfosit proliferation in vitro. Asian Journal of Pharmaceutical and Clinical Research 11(15): 85-88, doi:10.22159/ajpcr.2018.v11s3.30041

---

## [Editor Report · Decision Letter 2]

12 Jul 2021

Experimental and Molecular Predictions of the Adjuvanticity of Snail Mucin on Hepatitis B Vaccine in Albino Mice

PONE-D-21-02861R2

Dear Dr. Ezeorba,

We’re pleased to inform you that your manuscript has been judged scientifically suitable for publication and will be formally accepted for publication once it meets all outstanding technical requirements.

Kind regards,

Isabelle Chemin, PhD

Academic Editor

PLOS ONE
---

## [Editor Report · Acceptance letter]

15 Jul 2021

PONE-D-21-02861R2 

Experimental and Molecular Predictions of the Adjuvanticity of Snail Mucin on Hepatitis B Vaccine in Albino Mice 

Dear Dr. Ezeorba:

I'm pleased to inform you that your manuscript has been deemed suitable for publication in PLOS ONE. Congratulations! Your manuscript is now with our production department. 

Kind regards, 

on behalf of

Mrs Isabelle Chemin 

Academic Editor

PLOS ONE